Age differences in head motion and estimates of cortical morphology

Madan Christopher R. christopher.madan@nottingham.ac.uk
School of Psychology, University of Nottingham , Nottingham , United Kingdom
Comani Silvia
Electronic publication date: 2018 Jul 27
Publication date: 2018
Volume: 6
Electronic Location ID: e5176
Received 2018 Mar 2; Accepted 2018 Jun 16
Copyright: ©2018 Madan
Copyright year: 2018
Copyright holder: Madan
License: This is an open access article distributed under the terms of the Creative Commons Attribution License, which permits unrestricted use, distribution, reproduction and adaptation in any medium and for any purpose provided that it is properly attributed. For attribution, the original author(s), title, publication source (PeerJ) and either DOI or URL of the article must be cited.
License URL: https://creativecommons.org/licenses/by/4.0/

Keywords: Head motion, Cortical structure, Fractal dimensionality, Age, Cortical thickness, Gyrification, Cortical morphology, Movie watching, BMI

Funding: UK Biotechnology and Biological Sciences Research Council (BBSRC) BB/H008217/1 UK Medical Research Council (MRC) University of Cambridge CamCAN funding was provided by the UK Biotechnology and Biological Sciences Research Council (BBSRC) (BB/H008217/1), together with support from the UK Medical Research Council (MRC) and the University of Cambridge. The funders had no role in study design, data collection and analysis, decision to publish, or preparation of the manuscript.

==============================
Cortical morphology is known to differ with age, as measured by cortical thickness, fractal dimensionality, and gyrification. However, head motion during MRI scanning has been shown to influence estimates of cortical thickness as well as increase with age. Studies have also found task-related differences in head motion and relationships between body–mass index (BMI) and head motion. Here I replicated these prior findings, as well as several others, within a large, open-access dataset (Centre for Ageing and Neuroscience, CamCAN). This is a larger dataset than these results have been demonstrated previously, within a sample size of more than 600 adults across the adult lifespan. While replicating prior findings is important, demonstrating these key findings concurrently also provides an opportunity for additional related analyses: critically, I test for the influence of head motion on cortical fractal dimensionality and gyrification; effects were statistically significant in some cases, but small in magnitude.

Introduction

Head motion during the acquisition of magnetic resonance imaging (MRI) can lead to artifacts when estimating brain activity and structure. With functional MRI (fMRI), volumes are acquired relatively quickly–often every 1–3 s–allowing for the estimation and correction of head motion artifacts. Using innovative techniques such as prospective motion correction (Dosenbach et al., 2017; Federau & Gallichan, 2016; Maclaren et al., 2013; Stucht et al., 2015; Tisdall et al., 2016) and custom-designed, individualized head-cases (https://caseforge.co), effects of head motion can be attenuated. However, these solutions are not suitable for large studies of inter-individual differences in brain morphology where changes to the MRI scan sequence or custom-built equipment for each participant are often not practical. In the current study, I assessed relationships between age and body–mass index (BMI) on head motion, task-related differences in head motion, and the influence of head motion on estimates of cortical morphology. In light of these findings, many of which are replications, I propose a potential method for attenuating head motion during structural MRIs, as well as discuss limitations of this method.

Prior studies have demonstrated that older adults tend to have more head motion than younger adults (Andrews-Hanna et al., 2007; Chan et al., 2014; Savalia et al., 2017; Pardoe, Hiess & Kuzniecky, 2016). Unfortunately, other studies have also provided evidence that head motion can lead to lower cortical thickness estimates (Alexander-Bloch et al., 2016; Pardoe, Hiess & Kuzniecky, 2016; Reuter et al., 2015; Savalia et al., 2017), as such, age-related differences in cortical thickness (e.g., Fjell et al., 2009; McKay et al., 2014; Salat et al., 2004) may be exaggerated by age-related differences in head motion. In addition to age, obesity has also been associated with head motion (Beyer et al., 2017; Hodgson et al., 2017). In particular, these associations have been shown with respect to body–mass index (BMI; kg/m2), which is measured as body weight (in kg) divided by body height (in m) squared–despite the relatively coarse nature of BMI (e.g., does not differentiate between muscle vs. fat mass) (Diverse Populations Collaborative Group, 2005; Romero-Corral et al., 2008). Findings of relationships between obesity and cortical thickness have been mixed (Shaw et al., 2017; Shaw et al., 2018; Veit et al., 2014). More generally, head motion has been suggested to be a neurobiological trait–being both stable over time and heritable (Engelhardt et al., 2017; Hodgson et al., 2017; Zeng et al., 2014).

There is also evidence that fMRI tasks can differ in the degree of associated head motion (Alexander et al., 2017; Huijbers et al., 2017; Greene et al., 2018; Vanderwal et al., 2015; Wylie et al., 2014; Yuan et al., 2009). With this in mind, it may be beneficial to present participants with a task to attend to during structural scans, with the objective of decreasing head motion; typically structural scans are accompanied by the presentation of a blank screen or otherwise lack of instruction of attending to a visual stimulus.

Madan & Kensinger (2016) showed that a structural metric, fractal dimensionality (FD), may be more sensitive to age-related differences in cortical structure than cortical thickness (also see Madan & Kensinger, 2018). In a preliminary analysis to examine the influence of head motion on age-related differences in cortical fractal dimensionality, Madan & Kensinger (2016) showed qualitative evidence of age-related differences in fractal dimensionality in a small sample (N = 7) of post-mortem MRIs. However, as this sample was small and also less indicative of potential head motion effects in in vivo MR imaging, further work is necessary. To more directly test for the additive influence of head motion on estimates of cortical morphology, beyond aging, here I also tested for a relationship of fMRI-estimated head motion on cortical fractal dimensionality, as well as on mean cortical thickness. Additionally, as recent studies have found that gyrification also decreases with age (Cao et al., 2017; Hogstrom et al., 2013; Madan & Kensinger, 2016; Madan & Kensinger, 2018), it was also included in the analysis presented here. Test-retest reliability of estimates for these structural measures has recently been compared (Madan & Kensinger, 2017b), but robustness to head motion has yet to be assessed.

Using the rich, open-access dataset from Cambridge Centre for Ageing and Neuroscience (CamCAN) (Shafto et al., 2014; Taylor et al., 2017), here I sought to replicate these myriad of prior findings, as well as test for influences of head motion on fractal dimensionality and gyrification.

Methods

Dataset

Data used in the preparation of this work were obtained from the Cambridge Centre for Ageing and Neuroscience (CamCAN) repository, available at http://www.mrc-cbu.cam.ac.uk/datasets/camcan/ (Shafto et al., 2014; Taylor et al., 2017). The CamCAN dataset includes structural and functional MRI data for a sample of 648 adults across the adult lifespan (aged 18–88; Mean (SD) =54.2(18.5)). All participants were cognitively healthy (MMSE >24) and were free of any neurological or serious psychiatric conditions. See Shafto et al. (2014) for additional details about the sample inclusion and exclusion criteria.

A total of eight participants were excluded from further analyses due to problems with cortical reconstruction or gyrification estimation, yielding a final sample size of 640 adults (326 female, 314 male). Height and weight measurements were available for 559 of the 648 participants (280 female, 279 male), additionally allowing for the calculation of body–mass index (BMI) for this subset of participants (also see Ronan et al., 2016).

Structural measures are derived from a T1-weighted volume acquired using a 3 T Siemens Trio MRI scanner with an MPRAGE sequence. Scan parameters were as follows: TR  = 2,250 ms, TE  = 2.99 ms, flip angle  = 9°, voxel size  = 1 × 1 × 1 mm, GRAPPA  = 2, TI  = 900 ms. Head motion was primarily estimated from two fMRI scans, during rest and a movie-watching task. Both scans lasted for 8 min and 40 s (i.e., 520 s total). For the rest scan, participants were instructed to rest with their eyes closed. For the movie scan, participants watched and listened to condensed version of Alfred Hitchcock’s (1961) “Bang! You’re Dead” (Campbell et al., 2015; Hasson et al., 2008). Note that different scan sequences were used for both of these scans, with volumes collected every 1.970 s or 2.470 s for the rest and movie scans, respectively (see Taylor et al., 2017 for more details); both rest and movie scans had the same voxel size, 3 × 3 × 4.44 mm (32 axial slices, 3.7 mm thick, 0.74 mm gap).

Preprocessing of the structural MRI data

The T1-weighted structural MRIs were processed using FreeSurfer v6.0 (https://surfer.nmr.mgh.harvard.edu/) (Dale, Fischl & Sereno, 1999; Fischl, 2012; Fischl & Dale, 2000). Surface meshes and cortical thickness was estimated using the standard processing pipeline, i.e., recon-all, and no manual edits were made to the surfaces. Gyrification was calculated using FreeSurfer, as described in Schaer et al. (2012).

Fractal dimensionality (FD) is a measure of the complexity of a structure and has previously been shown to decrease in relation to aging for cortical (Madan & Kensinger, 2016; Madan & Kensinger, 2018) and subcortical (Madan & Kensinger, 2017a; Madan, 2018) structures and has been shown to have high test-retest reliability (Madan & Kensinger, 2017b). FD was calculated using the calcFD toolbox (http://cmadan.github.io/calcFD/) (Madan & Kensinger, 2016) using the dilation method and filled structures (denoted as FDf in prior studies). Briefly, FD measures the effective dimensionality of a structure by counting how many grid ‘boxes’ of a particular size are needed to contain a structure; these counts are then contrasted relative to the box sizes in log-space, yielding a scale-invariant measure of the complexity of a structure. This is mathematically calculated as FD =  − Δlog2(Count)∕Δlog2(Size), where Size was set to {1, 2, 4, 8, 16} (i.e., powers of 2, ranging from 0 to 4). To correct for the variability in FD estimates associated with the alignment of the box-grid with the structure, a dilation algorithm was used which instead relies on a 3D-convolution operation (convn in MATLAB) as this approach yields more reliable estimates of FD. This computational issue is described mathematical and demonstrated in simulations in Madan & Kensinger (2016), and empirically shown in Madan & Kensinger (2017b). See Madan & Kensinger (2016) and Madan & Kensinger (2018) for additional background on fractal dimensionality and its application to brain imaging data.

Estimates of head motion

Head motion was estimated using two approaches:

(1) Measured as the frame-wise displacement using the three translational and three rotational realignment parameters. Realignment parameters were included as part of the preprocessed fMRI data (Taylor et al., 2017), in the form of the rp_*.txt output generated by the SPM realignment procedure. Rotational displacements were converted from degrees to millimeters by calculating the displacement on the surface of a sphere with a radius of 50 mm (as in Power et al., 2012). Frame-wise displacement was substantially higher between volumes at the beginning of each scan run, so the first five volumes were excluded. This is the same approach to estimating head motion that is commonly used (e.g., Alexander-Bloch et al., 2016; Engelhardt et al., 2017; Power et al., 2012; Savalia et al., 2017).

(2) Estimated directly from the T1-weighted volume as ‘average edge strength’ (AES) (Aksoy et al., 2012; Zacà et al., in press). This approach measures the intensity of contrast at edges within an image. Higher AES values correspond to less motion, with image blurring yielding decreased tissue contrast. AES was calculated using the toolbox provided by Zacà et al. (in press), on the skull-stripped volumes generated as an intermediate stage of the FreeSurfer processing pipeline. AES is calculated on two-dimensional image planes and was performed on each plane orientation (axial, sagittal, and coronal).

Model comparison approach

Effects of head motion on estimates of cortical morphology (thickness, fractal dimensionality, and gyrification) were assessed using a hierarchical regression procedure using MATLAB. Age was first input, followed by BMI (both with and without age), followed by estimates of head motion from each fMRI scan and the related interaction term with age. In total, eight models were examined, as listed in Table 1. Model fitness was assessed using both R2 and ΔBIC.

Bayesian Information Criterion, BIC, is a model fitness index that includes a penalty based on the number of free parameters (Schwarz, 1978). Smaller BIC values correspond to better model fits. By convention, two models are considered equivalent if ΔBIC < 2 (Burnham & Anderson, 2004). As BIC values are based on the relevant dependent variable, ΔBIC values are reported relative to the best-performing model (i.e.,  ΔBIC = 0 for the best model considered).

Table 1 Variance explained and model fits of cortical measures by age, BMI, and head motion estimates. Note that R2 decreases after the inclusion of BMI as models 2 and 3 can only be calculated on a subset of participants (559 out of 640 participants) since height and weight information was not available for all participants.

		Thickness	FD	Gyrification	
Model	Predictors	R2	ΔBIC	R2	ΔBIC	R2	ΔBIC	
1	Age	.425	6.98	.497	3.15	.192	3.65	
2	BMI	.029	455.85	.028	805.25	.007	243.17	
3	Age + BMI	.425	168.82	.487	454.69	.183	140.23	
4	Age + Movement(Rest)	.429	10.01	.500	5.65	.192	10.07	
5	Age + Movement(Movie)	.437	0.00	.504	0.00	.194	8.44	
6	Age + Movement(Movie) + Age × Movement (Movie)	.427	11.64	.499	6.58	.205	0.00	
7	Age + AES(axial)	.443	0.23	.507	3.18	.194	14.83	
8	Age + AES(axial) + Age × AES(axial)	.428	17.58	.500	12.42	.208	3.76	

Results

fMRI-estimated head motion

As shown in Fig. 1, older adults head increased head motion relative to younger adults in both the rest and movie scans (rest: r(638) = .351, p < .001; movie: r(638) = .430, p < .001). Head motion was also greater in the rest scan than during the movie watching (t(639) = 23.35, p < .001, Cohen’s d = 0.99, Mdiff = 1.528 mm/min). Nonetheless, head motion was correlated between the fMRI scans [r(638) = .484, p < .001]. While this correlation between scans is expected, particularly since both were collected in the same MRI session, studies have provided evidence that head motion during scanning may be a trait (Engelhardt et al., 2017; Hodgson et al., 2017; Zeng et al., 2014). Moreover, this correlation provides additional evidence that motion during the fMRI scans is consistently larger in some individuals than others, suggesting it similarly affected the structural scans more for some individuals than others and appropriate to include as a predictor for the cortical morphology estimates.

Figure 1 Age-related differences in head motion.

Correlations between average head motion (mm/min) with age for the (A) rest and (B) movie fMRI scans, with (D–E) body–mass index (BMI), (C) between fMRI scans, and (F) between age and BMI. Head motion axes are log-10 scaled to better show inter-individual variability.

As expected based on prior literature (Beyer et al., 2017; Hodgson et al., 2017), head motion was also correlated with body–mass index (BMI) (rest: r(557) = .456, p < .001; movie: r(557) = .335, p < .001) (Fig. 1). While BMI was also correlated with age (r(557) = .274, p < .001), BMI-effects on head motion persisted after accounting for age differences (rest: rp(555) = .340, p < .001; movie: rp(555) = .249, p < .001).

While head motion was substantially lower in the movie condition than during rest, it was relatively stable over time (e.g., it does not tend to decrease over time). However, in the movie watching task, there is evidence of systematic stimuli-evoked increases in head motion (Fig. 2), e.g., around 280 s and 360 s. These periods of increased head motion correspond to events within the movie; in the first period, the boy is loading the real gun with bullets, the second, more prominent period is a suspenseful scene where it appears that the boy may accidentally shoot someone. Moreover, these events also correspond to fMRI differences in attentional control and inter-subject synchrony (see Campbell et al., 2015).

Figure 2 Averaged time-course of head motion for rest (red) and movie (blue) fMRI scans for young (A) and older adults (B).

Bands represent 95% confidence intervals.

Figure 3 Relationships between motion estimated from the structural volume using average edge strength (AES) in (A–C) different planes with age, (D) between planes, (E) BMI, (F) rest-fMRI estimated motion, (G) cortical thickness, and (H) fractal dimensionality.

T1-estimated head motion

Head motion was also estimated directly from the T1-weighted volume as the average edge strength (AES), following from Zacà et al. (in press); higher AES values correspond to less motion. Here I calculated AES for each plane orientation. AES in the axial and sagittal planes was moderately related to age (axial: r(639) = .493, p < .001; sagittal: r(639) = .525, p < .001) (Fig. 3); AES in the coronal was only weakly correlated with age (r(639) =  − .131, p < .001). AES in the axial and sagittal planes were strongly correlated with each other (r(639) = .702, p < .001).

Interestingly, AES was relatively not related to BMI (all |r|’s <.2). AES was also relatively unrelated to fMRI-estimated head motion (rest: r(639) = .112, p = .005; movie: r(639) = .148, p < .001). Thus, while AES is sensitive to an MR image property related to age, it seems to be distinct from fMRI-estimated head motion. A likely possibility is that AES here is detecting age-related differences in gray/white matter contrast ratio (GWR), as have been previously observed (Knight et al., 2016; Magnaldi et al., 1993; Salat et al., 2009). In contrast, the mechanism for the correlation between BMI and fMRI-estimated head motion is likely apparent–rather than real–head motion caused by respiratory chest motion producing susceptibility variations in the B0 field (Raj, Anderson & Gore, 2001; Van de Moortele et al., 2002; Van Gelderen et al., 2007).

Figure 4 Age- and BMI-related differences in the three cortical morphology measures examined here: (A, D) thickness, (B, E) fractal dimensionality, and (C, F) gyrification.

Cortical morphology

As shown in Fig. 4, mean cortical thickness significantly decreased with age (r(638) =  − .652, p < .001, −0.0432 mm/decade), as did fractal dimensionality (r(638) =  − .705, p < .001, −0.0097 FDf/decade) and gyrification (r(638) =  − .427, p < .001, −0.0372 GI/decade). All three slopes (change in metric per decade) are nearly identical to those first calculated by Madan & Kensinger (2016), as is the finding of higher age-related differences in fractal dimensionality and weaker differences in gyrification (also see Madan & Kensinger, 2018). However, it is also worth acknowledging that AES in the axial and sagittal planes were comparably correlated with age as gyrification. Effects of BMI on all three measures of cortical morphology were relatively weak (thickness: r(557) =  − .169, p < .001; fractal dimensionality: r(557) =  − .168, p < .001; gyrification r(557) =  − .083, p = .049).

Of particular interest, I examined the influence of head motion on the cortical morphology estimates. For all three measures, head motion explained only a small amount of additional variance beyond age, as shown in Table 1. Nonetheless, head motion from the movie scan did explain significant additional variance, as measured by ΔBIC, however, this only accounted for an additional 1% variance in the cortical morphology measures. In the model of cortical thickness including head motion from the movie scan (but not the interaction), age related changes corresponded to −0.0398 mm/decade, while head motion contributed −0.0135 mm/(mm/min).

Discussion

In the current study, I replicated several prior findings as well as tested for a few novel effects of head motion. First I outline the key findings of prior studies that were replicated here:

(1) Increased head motion in older adults (replicating Savalia et al., 2017; Pardoe, Hiess & Kuzniecky, 2016).

(2) BMI is correlated with fMRI-estimated head motion (replicating Beyer et al., 2017; Hodgson et al., 2017).

(3) Less head motion occurs when watching a movie than during rest (replicating Vanderwal et al., 2015; Huijbers et al., 2017).

(4) Head motion in different scans from the same individuals is correlated and indexes reliable inter-individual differences (replicating Zeng et al., 2014; Engelhardt et al., 2017; Hodgson et al., 2017).

(5) Cortical thickness decreases with age (replicating Fjell et al., 2009; Salat et al., 2004).

(6) Fractal dimensionality and gyrification also decrease with age (replicating Madan & Kensinger, 2016; Madan & Kensinger, 2018; Hogstrom et al., 2013).

(7) More head motion leads to lower estimates of cortical thickness (replicating Reuter et al., 2015; Savalia et al., 2017).

In addition to these replications, the new findings were:

(8) Head motion leads to nominally lower estimates of fractal dimensionality and gyrification.

(9) Head motion estimated from the structural volume itself (i.e., average edge strength [AES]) correlated with age, but not BMI.

(10) AES may be sensitive to gray/white matter contrast ratio (GWR).

(11) AES was only weakly related to fMRI-estimated head motion.

(12) Global cortical morphology is weakly related to BMI.

Likely most important, I found significantly more movement during resting state than watching a movie, but are quite correlated still (replicating the findings of Huijbers et al., 2017; Greene et al., 2018). Based on this evidence, I would recommend that participants be given movie-watching task during structural scans to reduce movement during these longer volume acquisitions and improve scan quality. Suggestions of potential systematic increases in head motion, however, suggest that less eventful movie content may be preferable for both maintaining participants’ attention and minimizing movement-based reactions (e.g., see Vanderwal et al., 2015). While this approach is not common, it has been used in some recent large-scale studies, such as the Human Connectome Project (HCP) (Marcus et al., 2013) and Adolescent Brain Cognitive Development (ABCD) study (Casey et al., in press), and has also been suggested and used elsewhere, particularly in MRI studies with children (Greene, Black & Schlaggar, 2016; De Bellis et al., 2001; Howell et al., in press; Overmeyer, 1996; Pliszka et al., 2006; Raschle et al., 2009; Theys, Wouters & Ghesquière, 2014; Von Rhein et al., 2015; Wu Nordahl et al., 2008). However, it is also important to consider the context that this movie watching would occur in. For instance, if the structural scan is followed by a resting-state fMRI scan, cognitive processes related to the movie watching will ‘spill over’ and influence patterns of brain activity in a subsequent rest period (e.g., Tambini & Davachi, 2013; Van Kesteren et al., 2010; Eryilmaz et al., 2011).

Estimates of cortical thickness were significantly influenced by head motion (replicating Savalia et al., 2017; Reuter et al., 2015), though the influence of this appeared to be relatively small. Effects of head motion on fractal dimensionality were also significant, but even smaller in magnitude, while head motion did not significantly influence estimates of gyrification. The results here also served as a replication age-related differences in fractal dimensionality and gyrification (Madan & Kensinger, 2016; Madan & Kensinger, 2018).

Interestingly, average edge strength (AES) did not correlate well with fMRI-estimated motion, but did correlate with age. This may be related to age-related differences in gray/white matter contrast ratio (GWR), as AES corresponds to the degree of tissue intensity contrast. This finding may be important when examining differences in AES between different samples (e.g., patients vs. controls).

While the results here are predominately replications of prior work, they nonetheless integrate the key findings of several papers through a single, open-access dataset, that also has a larger sample size than these previous studies. Moreover, these results serve as an example to highlight the benefits of open data sharing on improving our understanding of brain morphology (see Madan, 2017 for a detailed discussion).

Conclusion

Head motion influences estimates of cortical morphology, but can be attenuated by using an engaging task, such as movie watching, rather than merely instructing participants to rest. Decreasing head motion is particularly important when studying aging populations, where head motion is greater than for young adults, but considerations are necessary to see how this may ‘carry over’ and influence a subsequent scan, such as resting-state fMRI.

Supplemental Information

Supplemental Information 1 Derived brain morphology measures from CamCAN data

Each row corresponds to one participant. The “mvmt” columns correspond to estimated head motion in the rest and movie-watching fMRI scans. “ct”, “gyrif”, and “fd” correspond to estimated cortical morphology measures: cortical thickness, gyrification, and fractal dimensionality. The “aes” columns correspond to the AES measure in the axial, coronal, and sagittal planes, respectively. BMI is the body-mass index.

Click here for additional data file.

Data collection and sharing for this project was provided by the Cambridge Centre for Ageing and Neuroscience (CamCAN). I would like to thank Darren Price and Rogier Kievit for assistance with accessing the CamCAN data. I would also like to thank Jordan Theriault and Alexis Porter for insightful discussions.

Additional Information and Declarations

Competing Interests

Author Contributions

Data Availability

The authors declare there are no competing interests.

Christopher R. Madan conceived and designed the experiments, analyzed the data, contributed reagents/materials/analysis tools, prepared figures and/or tables, authored or reviewed drafts of the paper, approved the final draft.

The following information was supplied regarding data availability:

The unprocessed T1 structural data, head-motion regressors from the processed resting-state and movie-watching functional MRI data, along with demographic (age, sex) and physical (height, weight) data are available at Cam-CAN Data Portal

https://camcan-archive.mrc-cbu.cam.ac.uk/dataaccess/.

The measures derived in this article’s analysis are available here: Christopher R. Madan. (2018). Derived brain morphology measures from CamCAN data [Data set]. Zenodo. http://doi.org/10.5281/zenodo.1258016.

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
