# Peer review of "Age differences in head motion and estimates of cortical morphology"

_PeerJ, doi:10.7717/peerj.5176_

## Round 0.1 · original submission · Major Revisions

· Academic Editor

Major Revisions

I strongly suggest that you address all comments by both reviewers befor submitting your revised manuscript, as I am convinced that this will increase the value of your work.

Reviewer 1 ·

Basic reporting

• Major points.

1. Generally clear, professorial language.

2. Clear progression, structure and background.

3. Well formatted and clear.

• Minor points

1. Slight typo on line 16 – perhaps should be 'also differs'

2. Line 16 ‘Here I…’ is a slightly awkward sentence and could be clearer, e.g. ‘within…’ to ‘with a larger dataset than has been demonstrated previously’.

3. Consider referencing the Hitchcock film used fully (e.g. https://learn.solent.ac.uk/mod/book/view.php?id=3381&chapterid=3455).

4. Consider referencing previous literature regarding ‘Models with differences in BIC>2 were considered…’ (line 106-107).

Experimental design

• Major points

1. Well defined and a meaningful question.

2. Conducted to a generally high technical and moral standard.

3. Generally, sufficiently detailed.

4. Is there a male vs. female effect? Some basic demographic details like sex frequencies, average age, stroke frequency etc. for the sample would be helpful. If these are stated in other papers (for CamCAN), please explicate that.

• Minor points

1. Perhaps IQ or personality metrics predict head motion at rest/movie-condition, or the amount of improvement from rest->film, independent of older age. You could consider this analytically or note for future research (if you think this might be a valid factor; only a suggestion, and ditto this for any possible sex effect as suggested above).

Validity of the findings

1. Very important, easily-applied amendments for human brain imaging research (i.e. that films should be shown during structural scans). I believe the data are robust, statistically sound and this cautious conclusion is reasonable in that context.

2. A generally commendable paper which conducts detailed analysis of a well-outlined problem, and links conclusions to robust data and questions.

Comments for the author

A good, clear paper.

·

Basic reporting

For the most part the paper is well written in proper English. However, I did not find the background information presented in the introduction sufficient to fully justify the work and to understand the importance of the findings. Without further elaboration, it is not clear what problem or issue this work addresses. Although the work serves as a replication of some of the authors prior findings, the results are completely predictable and there is no clear indication why such a replication is either necessary or helpful.

I think this overall weakness is reflected in the discussion which highlights only that this work found the same correlations previously reported by the author.

In my view, the introduction needs substantial elaboration to understand how this work contributes to our understanding of how head motion influences estimates of cortical thickness in aging.

More minor potential edits:
In the opening paragraph, it appears that the author means to say that head motion leads to lower estimated cortical thickness – but the language of “decreased estimates of cortical thickness” on line 34 is ambiguous and should be edited.

On line 36 the author uses the term “Taken together” to summarize studies that have not been discussed (or at least adequately discussed).

Experimental design

The methods lack detail about the data and how it was analyzed. Although the data come from a database and the techniques have been used and reported elsewhere, it is necessary that this paper stand on its own without the needs to reference a handful of other works to grasp the implementation.

Please provide information about the MRI machine, fMRI pulse sequence, number of slices, preprocessing on fMRI data, etc.

A more detailed description of FD should be presented in the paper rather than referring readers to other papers, particularly because it appears to be an approach developed by the author and is not widely known or adopted.

Describe the measure of frame-wise displacement and why this measure was used instead of other measures of head motion.

Section 2.4 Model comparison approach is under-described and contains no citations for referencing the explanatory tools or their application. In particular, describe your hierarchical regression procedure including what software was used. Also provide more detail on model construction implementation. It appears that age and head motion may be at different levels of the model, but the description is somewhat vague. Describe Bayesian Information Criterion (including writing it out in words before using the abbreviation) in more detail including how it was applied to the data in the study. The use of this measure implies the evaluation of different models. Be explicit about how many models were tested, how each was constructed, and which model comparisons were made.

Validity of the findings

The author makes an argument that the fMRI sessions are an appropriate predictor of motion during structural scans based on a correlation between the two fMRI sessions. I think it would be useful to discuss possible limitations of this assumption in the discussion section.

Without further analysis and evidence, there is no justification for the observation beginning on line 121 for systematic stimuli evoked increases in head motion.

It seems that with the available data there is an opportunity to provide more detailed and potential more interesting data about head motions effect on thickness estimates in different regions of the cortex, particularly because thickness estimates are not influences uniformly across the brain.

The delta BIC in table 1 is difficult to interpret because it is not clear what models are being compared. Moreover, if a delta BIC of greater than 2 is considered significant, why is only the Age + Motion(Movie) condition considered as containing additional explanatory power when 8 of the reported 12 delta BIC measures exceed a value of 2?

It would be helpful to see the BIC for each model in addition to their differences.

---

## Round 0.2 · accepted · Accept

· Academic Editor

Accept

As indicated by reviewer 2, there are some small typographical errors in the revised manuscript that need to be amended before publication. I think that you could carefully check your manuscript before publication.

·

Basic reporting

no comment

Experimental design

no comment

Validity of the findings

no comment

Comments for the author

The author has addressed my concerns, particularly with respect to the methods section of the paper. Although I would still like to see a more thoughtful discussion of the findings and at least some consideration of the source for the relationships (or lack thereof) reported in this work, I also recognize that PeerJ is concerned more with the findings than the scholarly impact. As such I feel the paper provides enough analysis and novel material to move forward to publication.

Note that there are some small typographical errors in the paper that I recommend the author correct before publication.